# A Reliable Delivery Logistics System Based on the Collaboration of UAVs and Vehicles

Hanxue Li [1], Shuaiqi Zhu [2], Amr Tolba [3] , Ziyi Liu [1] and Wu Wen [1,*]

1    School of Communication and Information Engineering, Chongqing University of Posts and Telecommunications, Chongqing 400065, China
2    School of Software, Dalian University of Technology, Dalian 116024, China
3    Department of Computer Science, Community College, King Saud University, Riyadh 11437, Saudi Arabia
*    Correspondence: wenwu@cqupt.edu.cn

**Abstract:** In recent years, land–air collaborative logistics delivery is a promising distribution method. It combines the flexibility of unmanned aerial vehicles (UAVs) with the high payload capacity of vehicles, expanding the service range of UAVs while reducing carbon emissions. However, most existing research has focused on path planning and resource allocation for either UAVs or vehicles alone. Therefore, to address the shortcomings of the current research, this paper proposes an intelligent land–air collaboration delivery algorithm for trajectory optimization and resource scheduling. Subsequently, this paper develops a land–air collaboration reliable delivery logistics distribution system, showcasing the driving routes of vehicles and UAVs. Meanwhile, the mode of UAV–vehicle collaboration not only saves operating costs compared to traditional logistics delivery but also achieves energy conservation and emission reduction goals. During the specific design and implementation process of this platform, blockchain technology is integrated into the logistics delivery service to ensure data security and prevent tampering, making the system more efficient and reliable. Finally, through testing and verification of the system's functionalities, its completeness is demonstrated.

**Keywords:** blockchain; UAV–vehicle collaboration; reliable delivery; logistics distribution

## 1. Introduction

In the past few decades, UAVs have gradually emerged in military operations, public safety, logistics, and delivery. With the prevalence of e-commerce, parcel delivery has become a crucial aspect of the logistics industry. Traditional parcel delivery is predominantly carried out through ground transportation. However, with the rise in labor costs and technological advancements, the use of UAVs in logistics and delivery is becoming increasingly attractive [1]. Limiting the negative effects of transport is an important goal of smart mobility in many regions. The main aspects of these activities involve shifting transport to the least polluting and most efficient modes of transport, using more sustainable transport technologies and infrastructure, and ensuring that transport prices fully reflect the negative environmental and health impacts [2]. Furthermore, an effective and real-time traffic information network is highly important as it can reduce traffic volume and costs by decreasing fuel consumption and saving time for drivers to reach their destinations [3]. Since the emergence of the COVID-19 pandemic, UAV delivery services have become more popular in the courier industry compared to traditional delivery methods. Furthermore, UAVs offer advantages such as low cost, lightweight, high maneuverability, and environmental friendliness [4,5]. Their high flexibility allows them to provide communication services to ground users [6]. While UAV delivery has notable advantages over traditional ground-based methods, it still faces challenges in terms of reliability and limited service coverage. UAVs alone struggle to efficiently achieve widespread logistics distribution. By combining

UAVs with ground vehicles and leveraging their unique features and advantages, it is possible to optimize the allocation of parcels, maximize service quality, and reduce costs.

With the rapid development of UAV technology, the associated security issues have gradually gained attention [7,8]. Taking the application in the logistics industry as an example, a group of UAVs needs to complete package delivery tasks in a specific area. During the mission, they should communicate with each other, allocate tasks, and plan their routes. Clearly, there is a need for security mechanisms to protect this communication from malicious attacks, prevent location tracking of packages, and prevent tampering with address information as these could result in significant economic losses for customers [7]. Blockchain is a promising technology, with characteristics such as anonymity, immutability, and traceability [9–12], making it a subject of extensive study for security and privacy issues in areas such as UAVs, the internet of things, and mobile networks [13]. Additionally, blockchain can record the flight paths of UAVs, delivery status, and cargo information, ensuring data security and privacy [14].

To enhance the efficiency and ensure the security of delivery services, leveraging UAVs, vehicles, and blockchain technology for proper planning and scheduling has become an effective method for achieving efficient and reliable logistics delivery services. The collaborative transportation model between UAVs and ground vehicles not only helps to reduce costs but also contributes to energy conservation and emission reduction. In recent research, wireless power transfer and mobile edge computing [15] have introduced new solutions for powering UAVs, aiming to improve their endurance. In the fifth-generation networks, one of the fundamental objectives is to increase the coverage and capacity of cellular networks by deploying additional base stations [16]. Although the future cellular communication network holds the potential to offer extensive connectivity among a large number of users, the existing cellular network is anticipated to fall short of meeting these demands. UAVs have emerged as a promising solution to address these requirements, acting as base stations [17]. Due to the mobility of UAVs [18], their base stations can hover or fly to various locations in the air, expanding the wireless coverage area and providing support for a large number of connections [19]. Compared to a single-vehicle service approach, the collaborative transportation of UAVs and vehicles offers significant advantages. Excessive vehicles can lead to traffic congestion, accidents, and road failures [20,21], while UAVs are not affected by complex ground conditions, avoiding issues such as road congestion, traffic control, and difficulties in reaching areas with challenging road conditions [22]. Therefore, in response to the current energy conservation and emission reduction challenges in China, land–aerial collaborative logistics delivery emerges as a promising distribution method. It can significantly reduce cost and package delivery time for service providers. By combining UAVs with blockchain technology, a more secure, transparent, and efficient logistics delivery process can be achieved. However, the joint trajectory optimization and resource allocation of UAVs and vehicles is a complex research area. Despite existing research focusing on trajectory optimization for UAVs and vehicles, limited research addresses reliable logistics delivery services supported by the combination of blockchain and UAVs.

- We design and develop a reliable delivery logistics distribution system based on the collaboration between UAVs and vehicles. This system has a positive impact on the development of logistics and distribution services while also visualizing the travel routes of vehicles and UAVs.
- The platform enriches the modes of logistics delivery and vehicle route planning, not only expanding the flight range of UAVs but also saving travel costs compared to traditional logistics delivery methods. Additionally, it reduces carbon emissions in the logistics process.
- Through the automatic execution and programmability of smart contracts, fast delivery confirmation, payment settlement, and service evaluation are achieved, simplifying the delivery process and reducing human errors.

- The integration of blockchain, UAVs, and vehicles in a reliable delivery logistics system provides higher efficiency, reliability, and sustainability to logistics delivery services.

The rest of this paper is organized as follows. Section 2 reviews related work. The system design is introduced in Section 3. System implementation is described in Section 4. Section 5 covers system testing and discussion, and Section 6 concludes this paper.

## 2. Related Works

With the establishment of smart cities and the development of network technology, intelligent transportation and vehicle logistics services are becoming ubiquitous in people's daily lives [23], bringing us many conveniences. The development of smart mobility plans requires specialized and context-specific policies to meet the needs and interests of various stakeholders [24]. Additionally, the intelligent transportation system aims to guide, accomplish, and manage various types of transportation systems [25]. However, the increasing number of services and applications in vehicular networks requires significant computing resources and real-time feedback. This poses challenges to vehicles with limited resources and centralized traffic management mechanisms [26]. Additionally, the rising costs of vehicle transportation and the accompanying increase in carbon dioxide emissions need to be addressed to reach a carbon peak as early as possible. Consequently, the growing energy consumption and environmental pollution present severe challenges to the sustainable development of cities [27]. As the third-largest source of carbon dioxide emissions, accelerating green, low-carbon, and sustainable development, as well as the rational scheduling of traffic vehicles and comprehensive use of traffic network resources, are important directions for future development in China.

In recent years, with the emergence of "contactless delivery" models, the traditional express service industry is undergoing significant transformations. To address China's current challenges in energy conservation and emission reduction, contactless delivery service models, and the "last-mile" delivery problem, a new logistics delivery model has gradually emerged, which involves the collaborative delivery of UAVs and vehicles. However, the existing research mostly focuses on resource allocation for either UAVs or vehicles separately. Additionally, the security of data in delivery services may be compromised, such as the risks of data tampering or information leakage. In response to the above phenomenon, as shown in Table 1, we summarize several major papers for comparison. To tackle these issues, the development of blockchain and mobile edge computing technologies has brought significant potential for efficient utilization of user data [28,29], promoting data security, privacy protection, and improving data processing efficiency. In a recent study, Liu et al. [30] designed a blockchain-based hierarchical security threat assessment framework that utilizes edge computing and blockchain technology to share data in vehicular networks.

For the study of UAV delivery services, Jeong et al. [31] propose a UAV delivery logistics system that considers the energy consumption of UAV clusters and models the problem as mixed integer linear programming to optimize the flight path of UAVs. Zhao et al. [32] study the trajectory optimization problem of UAVs in the context of energy conservation, model the trajectory problem of UAVs as two coupled multi-intelligent stochastic games, and decouple the game problem by using distributed RL methods. Huang et al. [33] study the dynamic task scheduling problem based on UAVs, considering the random task arrival time and delivery rate. They propose a heuristic event scheduling framework to minimize delay time. Sawadsitang et al. [1] consider the uncertainty and delivery cost of UAV package delivery. They employ stochastic programming techniques to formulate the optimization and planning of joint ground and aerial delivery services and make decisions at different stages. Considering multi-user computing offload and edge server deployment, Ning et al. [34] study the UAV-supported multi-access edge computing network to minimize low computing costs in dynamic environments. Das et al. [35] research the mechanism of multi-vehicle cooperative work with multiple UAVs. They combine it with delivery trucks serving as mobile launch and charging stations for the UAVs, transporting packages

during their journey. Lee et al. [36] employ dynamic programming methods to ensure that UAVs achieve their targets with the minimum energy consumption, considering flight distance and time to formulate optimal control strategies.

In research on vehicle logistics delivery services, Zhang et al. [37] propose a vehicle-oriented dynamic ridesharing delivery system, where blockchain is utilized to construct the decentralized structure. In the framework of blockchain, the system has greater potential to facilitate carpooling profits and transaction security. Zhang et al. [38] simultaneously consider carpooling and package delivery services. They design an intelligent system for autonomous vehicles and solve the joint optimization problem using mixed-integer linear programming and the Lagrangian dual decomposition method. However, due to the increasing traffic volume in smart cities, mobile network operators struggle with allocating limited resources for collecting and processing real-time traffic information. The recent developments in edge computing and content caching in wireless networks enable intelligent transportation systems to provide high-quality services to vehicles [39]. Ning et al. [40] integrate mobile cloud computing and mobile edge computing and design a resource allocation algorithm to achieve efficient offloading decisions. Additionally, Wang et al. [41] establish a "last-mile" parcel delivery system and allocate delivery services based on the surface map. Yu et al. [42] propose an autonomous driving vehicle logistics system that utilizes autonomous vehicles for joint routing and charging. For the resource allocation problem of vehicles, the combination of artificial intelligence and edge computing enables flexible resource scheduling in vehicular networks [43].

The combination of blockchain technology and interdisciplinary research innovations provides opportunities for changes in many industries. Today, the reliable delivery logistics system based on the synergy of UAVs and vehicles provides advantages, such as high efficiency, low cost, sustainability, and safety to logistics delivery services. Blockchain technology enables real-time sharing and transparency regarding logistics information, allowing each stage of the logistics process to be recorded and traced. The allocation of resources between vehicles and UAVs reduces logistics time, transportation costs, traffic congestion, and carbon emissions, driving the sustainable development of the logistics industry. Ultimately, the combination of blockchain and UAVs provides new opportunities for innovation in the logistics delivery field and is expected to generate broad societal benefits.

**Table 1.** Comparison of major papers. ("$\sqrt{}$" if the solution satisfies the property, "$\times$" if not).

| Ref. | Focus | Main Contribution | Data Security | Land–Air Collaboration | Last-Mile Delivery |
|---|---|---|---|---|---|
| [35] | Drones and trucks for collaborative delivery | Propose a novel mechanism that synchronizes drones and delivery trucks. | $\times$ | $\sqrt{}$ | $\sqrt{}$ |
| [41] | Last-mile parcel delivery | Design last-mile package delivery system for smart car travel sharing. | $\times$ | $\times$ | $\sqrt{}$ |
| [44] | Drone path planning | Propose a new mode for drone parcel delivery that explores the public transportation network. | $\times$ | $\sqrt{}$ | $\times$ |
| Our work | Reliable parcel delivery through UAVs and vehicles collaboration | Develop a land–air collaborative reliable delivery logistics and delivery system. | $\sqrt{}$ | $\sqrt{}$ | $\sqrt{}$ |

## 3. System Design

### 3.1. System Requirements Analysis

Requirements analysis is an indispensable and crucial stage in the system development process. A good and adequate requirements analysis will lay a solid foundation for the subsequent stages of software development and effectively avoid unnecessary deviations during system development and maintenance. This subsection focuses on how the land–air collaborative logistics delivery platform can display different factors in the algorithm (such as network topology, the ratio of light to heavy packages, and maximum power of UAVs)

through interactive interfaces, how to provide real-time display of iteration-related data during algorithm execution, the variable process between algorithms under the influence of different metrics, and the visualization display of the system to introduce the requirements.

3.1.1. Overview of System Requirements

The process of using this system is as follows: the user can initialize the algorithm running parameters by selecting the network topology unit size, user demand distribution, and various parameters of the UAV through interactive buttons. After confirming the settings, the system starts the algorithm execution and displays the changes in the iterative process of the objective function during the algorithm execution. Simultaneously, it runs different comparison algorithms to display their trends under different modules of the system. After the algorithm execution, the path-planning module presents the optimal routes for UAVs and vehicles and shows the various performances of the algorithms discussed in this paper in other visualization modules.

Based on the above process, this system needs to meet the following requirements. Therefore, the overall use case diagram of this system is shown in Figure 1.

1.  Topology Parameter Management: The system displays the current network topology in detail and sets it up interactively. Among them, it mainly includes the display of network topology parameters, user quantity setting, UAVs parameter setting, comparative algorithm selection, and package distribution.
2.  Travel Route Display: It primarily includes the distribution of user nodes, the initial topology of the map, and the UAVs' and vehicles' driving routes. Before algorithm execution, the route display module mainly shows the distribution of users and their demands. During the execution process, the system will dynamically display the values of each iteration and visually present the driving routes of UAVs and vehicles generated by the algorithm.
3.  Algorithm Execution and Route Planning: The system calls the user allocation algorithm, fast iteration algorithm, and action selection algorithm to generate the final route and runs the user-selected comparison algorithm. Among them, the network topology and user demands are used as inputs for the user assignment algorithm. The rapid iteration algorithm makes incremental improvements to the current route during each iteration, primarily by swiftly optimizing the objective value using operators. Meanwhile, the action selection algorithm can intelligently choose suitable operators to enhance the current performance.
4.  Run Result Display: The system displays the results in tables according to the selected number of algorithms and provides a comparative analysis of the running results. After the algorithm execution is completed, the results of multiple algorithms will be actively pushed to the frontend for display. The frontend page will primarily showcase the proposed algorithms in this paper and their corresponding comparative algorithms.
5.  Target Result Display: The system displays the target result according to the initialization content set by the user and presents the changes in results through trends. This section mainly presents the trend in optimization objective values for the current algorithm and comparative algorithms under different network topology and UAV parameter conditions.
6.  Service Capability Display: It primarily calculates the capabilities of UAV service to each user based on the output paths of the current algorithm. By comparing the UAV service capabilities between the current algorithm and the comparative algorithms, it validates the advantages of the proposed algorithm in UAV scheduling. In addition, the system calculates and displays the service capacity variation in UAVs during the service process so that users can observe visually.
7.  System Performance Display: It primarily displays real-time updates based on the objective function values generated during each iteration of the algorithm execution. It presents the data in the form of a line chart, allowing users to observe the system's

stability and robustness. Additionally, this module provides responsive prompts in case of multiple consecutive iterations with unchanged values.

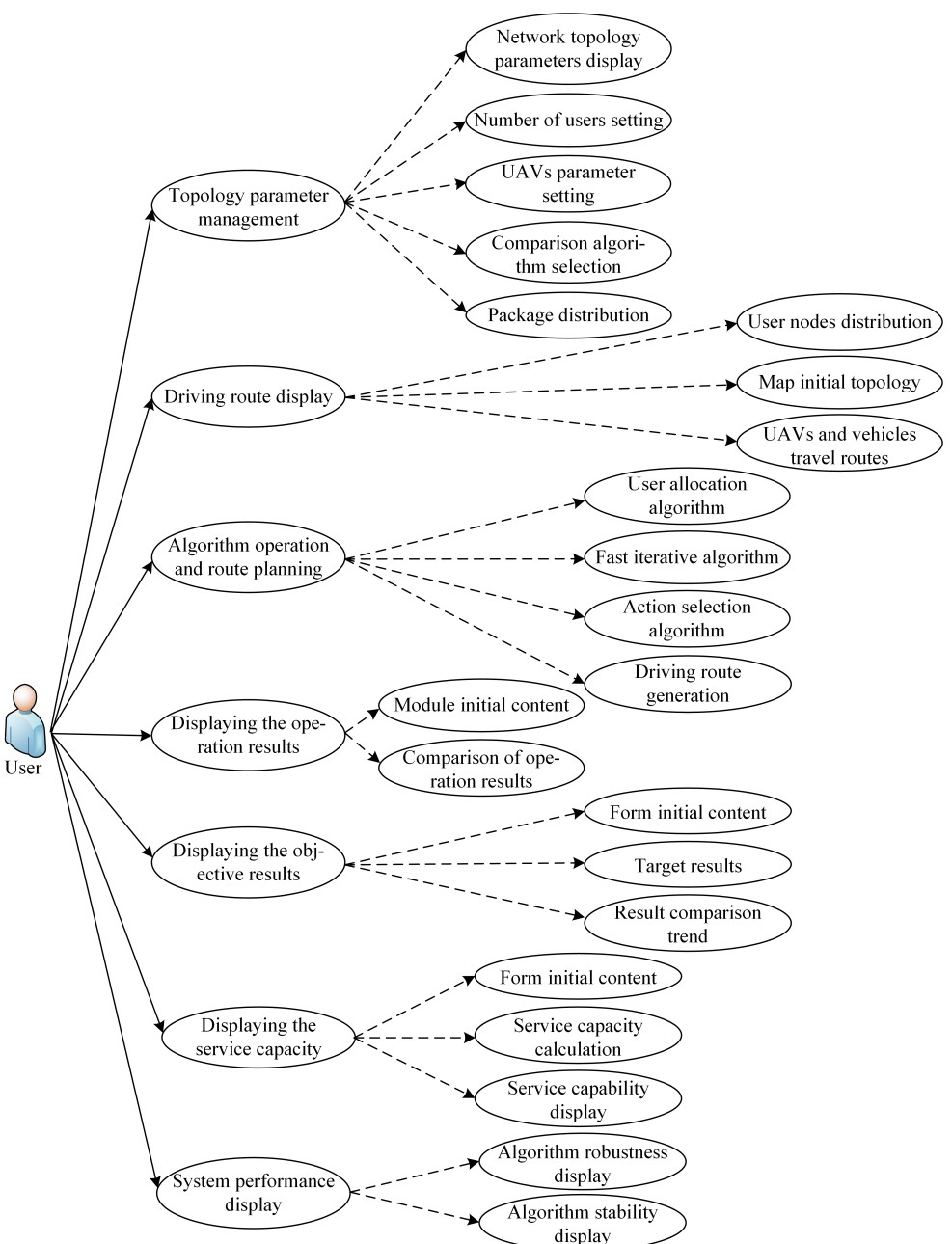

**Figure 1.** System general use case diagram.

### 3.1.2. System Goals

The objective of this system is to empower users to independently select network topology parameters and user distribution and configure various parameters related to UAVs. During the algorithm execution, the system provides a real-time display of the iteration process and shows the operation results of the algorithm through visualization. Additionally, the system is able to reasonably analyze the comparison results of different algorithms and show the trend changes of the algorithms. Based on these primary objectives, the system mainly addresses the following problems: the selection of network topology parameters, algorithm execution and visualization of UAV–vehicle paths, and visualization of algorithm metrics. In other words:

1.  Topology Parameter Selection: To validate the performance of the algorithms in different delivery scenarios, the system incorporates a topology parameter design module. Users can set different delivery scenarios using interactive buttons, allowing the evaluation of algorithm performance from various perspectives;
2.  Algorithm operation and UAV–vehicle path display: During the algorithm execution, it involves numerous iterations and associated values. Therefore, the system provides a real-time display of the values for each iteration. This allows the system users to observe the development trend of the algorithm;
3.  Visualization of Algorithm Metrics: Considering that the operation results of the algorithm in this paper and the comparison algorithm may be different, the system incorporates a visualization module to display data from various dimensions. This includes topology display, user node location display, route display, algorithm optimization target indicator display, and algorithm stability metric visualization. Through line charts, tables, bar graphs, and other visual forms, the system dynamically presents real-time updates of the current metrics, analyzes the strengths and weaknesses of the algorithms, and assists in explaining relevant algorithm concepts.

### 3.1.3. Overview of Functional Requirements

The integrated land–air logistics delivery platform requires the configuration of parameters for the current network, recording and displaying objective values during algorithm execution, and visualizing results, including operational outcomes, trends in UAV service capacity, algorithm stability, and robustness upon algorithm completion. Therefore, the main functionalities of this system can be categorized into three parts: network parameter configuration, algorithm execution tracking, and algorithm metric display.

(1)　Network Topology Parameter Setting Function

Module Design Objectives: The system displays the current network topology information to the user and provides an interface for the user to modify topology parameters, along with a button to trigger algorithm execution.

Specific Requirements Function: Users can initialize and set the current topology before running the algorithm. When the parameters are set, the algorithm can be run within the system. Network topology initialization mainly encompasses three components: setting the network topology size, setting UAV parameters (maximum payload, maximum power), and selecting comparative algorithms. The system provides four comparison algorithms alongside the algorithm presented in this paper. If the user does not select any parameters, the system will run with default topology parameters. The use case diagram for the network topology parameter setting functionality is shown in Figure 2.

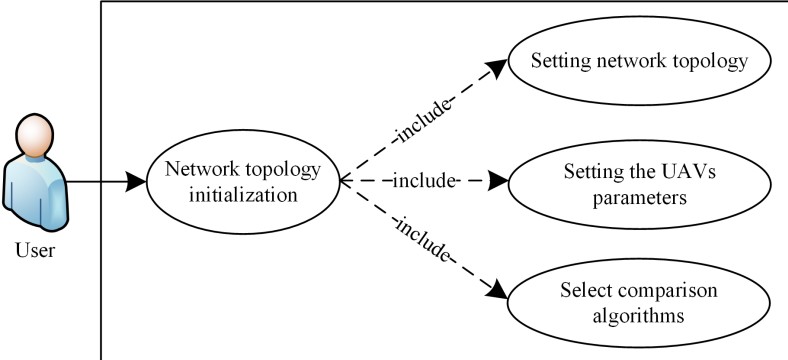

**Figure 2.** Use case diagram of set network topology parameters.

(2)　Algorithm Running Process Recording and Displaying Value Function

Module Design Goal: The system tracks and displays the optimization target values in real time during the algorithm iterations, providing users with real-time progress of the algorithm execution. This facilitates users to observe the algorithm running process.

Specific Requirement Function: Use a case diagram of the algorithm operation process record function as shown in Figure 3. During the execution of the system algorithm, it will first display the current network topology and the coordinate position of the user distribution. Throughout the execution process, the algorithm applies various operators to improve the current solution. Therefore, the system captures the values at each round of the algorithm and synchronously displays them, presenting the relationship between the number of iterations and the changes in the objective value using a trend graph. As this paper incorporates perturbation operators to monitor the evolution of the solution over multiple iterations, if the solution remains unchanged for multiple iterations, a perturbation controller is employed to modify the current solution. The system also marks cases where the values remain unchanged for multiple rounds, prompting the user that the system uses the perturbation controller at that point.

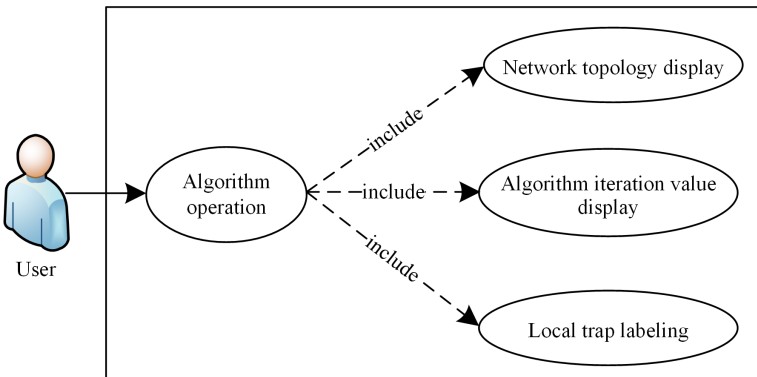

**Figure 3.** Use case diagram of algorithm operation process record.

(3) Multi-algorithm Metrics Display Function

Module Design Goal: The system visualizes from multiple perspectives, such as route driving trajectory, target result display, service capability, and algorithm performance display. These visualizations enable users to compare routes and performance trends in detail.

Specific Requirement Function: Use case diagram of the multi-algorithm index display as shown in Figure 4. After the algorithm finishes running, it will output the UAV flight route and vehicle driving route obtained by this algorithm. The system needs to display the routes on the network topology. Furthermore, it should calculate the UAV service capacity for each user node based on the route results of this algorithm and the comparison algorithms and visualize it in the form of a table. The system should also display the robustness, runtime, and numerical comparison results on the frontend page.

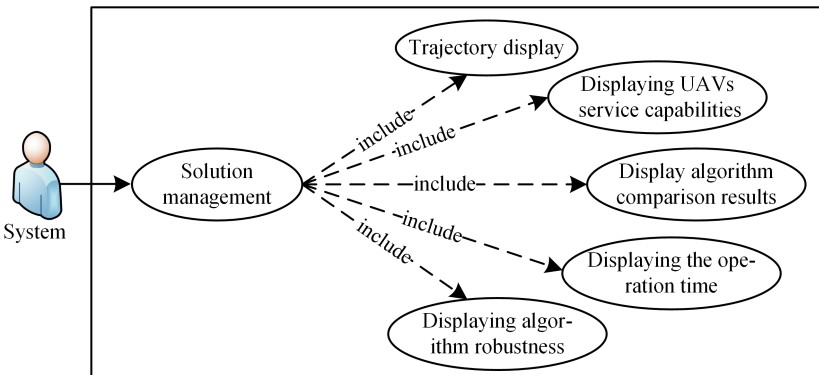

**Figure 4.** Use case diagram of multi-algorithm indicator display.

3.1.4. Overview of Non-Functional Requirements

Non-functional requirements analysis is a crucial part of system requirements analysis. Functional requirements define the specific functions of the system that are directly related to the users, while non-functional requirements ensure the overall characteristics of the system, guaranteeing its proper operation and maintenance. Non-functional requirements provide assurance for the smooth operation and maintenance of the system. In the following, non-functional requirements are analyzed based on considerations of the accuracy of information display, real-time data acquisition during the algorithmic process, and the readability of visualized values.

(1)　System Interface Simplicity Requirements

Firstly, as a visualization system, this system needs to have a clean and readable interface, showcasing different algorithm performances and metrics through various modules. It should employ animations, module interconnections, and user–system interactions to allow users to easily understand the performance differences and advantages between the current algorithm and comparison algorithms by adjusting a few parameters. Simultaneously, different display methods, such as tables, bar charts, line graphs, and trend charts, are chosen to display according to the characteristics of different dimensions.

Additionally, different evaluation criteria are used for various algorithms across diverse metrics. Therefore, it is necessary to provide appropriate information prompts during the pre-run, mid-run, and post-run stages of the system, for example, labeling the node positions for perturbation controller invocation in the robustness module, defining the service capability in the UAV service capability display module, and providing prompts during the topology configuration process to differentiate and define single and multiple objectives, facilitating user understanding and comparison.

(2)　Real-time System Operation Requirements

One of the functionalities of this system is to display the iteration results of the algorithm on a trend chart during each iteration round. Therefore, in each iteration process, the system needs to obtain the objective function value from the algorithm and dynamically display the corresponding data on the frontend trend chart. This non-functional requirement requires real-time communication between the frontend and backend, where the backend initiates communication and the frontend receives data and associates them with the iteration round for display. Similarly, when the backend calls the algorithm and completes the computation, it needs to send the results back to the frontend, and the frontend should listen for the returned data and invoke the corresponding callback function to promptly display the results.

(3)　System Display Accuracy Requirements

Due to the requirement of statistical analysis and display of different algorithms under various performance metrics, the system complexity in terms of interface interaction is increased. Therefore, it is necessary to design different data structures for each module and differentiate the algorithms involved in each module. This allows the display of different running results in their corresponding modules, thereby fulfilling the requirement of precise visualization in the system.

(4)　Overall System Stability Requirements

In order to handle potential issues, such as system crashes, frontend–backend interaction errors, and frontend display errors during different algorithm executions and recording, it is necessary to implement a comprehensive error interception and system logging mechanism to ensure the stability of the system during operation. Additionally, appropriate solutions should be implemented to address any issues that may arise, enabling timely invocation and emergency handling when problems occur, in order to ensure a smooth user experience and comfortable usage.

(5)   System Module Scalability Requirements

The system currently consists of multiple modules, but it should consider future scenarios in which users may demand the display of additional dimensions of the algorithm or the interactive adjustment of algorithm parameters. Therefore, in the system development process, it is necessary to encapsulate the different modules of the system, optimize module functionality using techniques like slots and component value passing, and prioritize the reusability and encapsulation of modules to ensure the system's scalability in usage and future development.

### 3.2. System Outline Design

### 3.2.1. System Overall Architecture Design

Based on the system requirements analysis in the previous section, this system is designed using a model view controller network architecture with high scalability and stability. The system adopts a layered design approach, dividing it into five layers: the presentation layer, data transmission layer, backend services layer, business services layer, and logic layer. Each of these layers corresponds to specific functionalities and is responsible for their respective business and functions. The specific system software architectural diagram is shown in Figure 5.

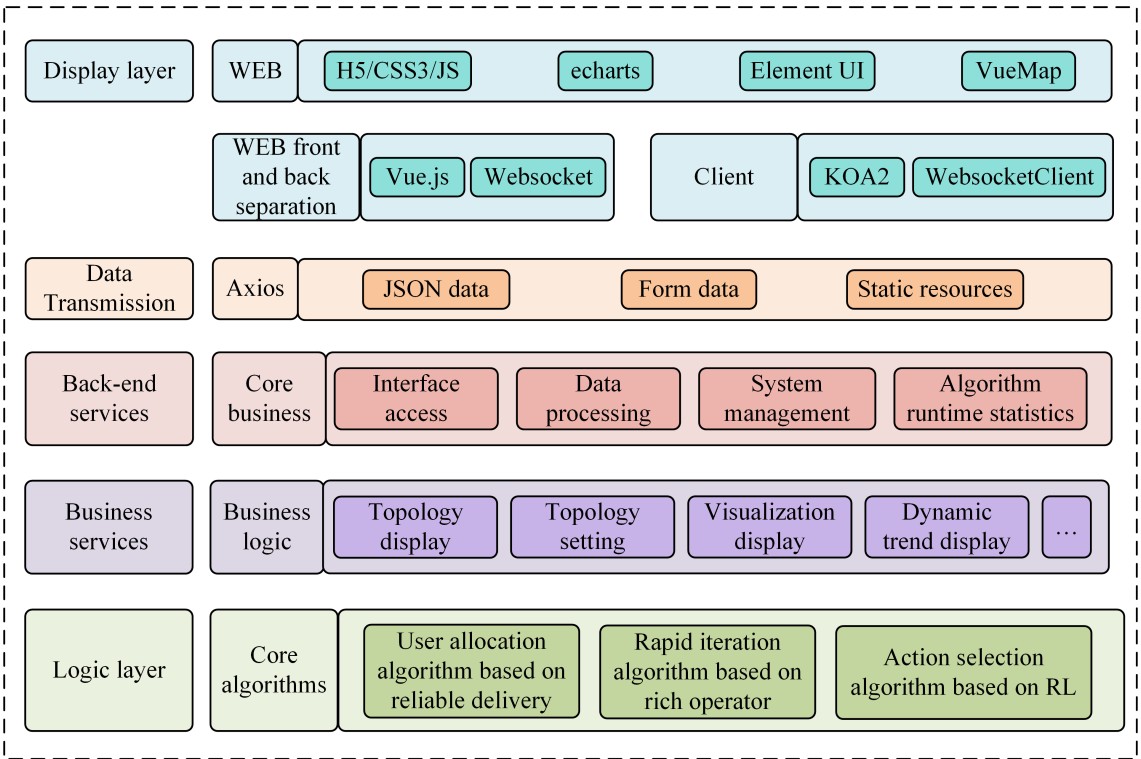

**Figure 5.** Software architecture mode diagram of the system.

In the presentation layer, the primary responsibility lies in frontend visualization functionality. Vue.js is used as the frontend framework, and visual components such as Echarts, VueMap, and ElementUI are integrated for development. H5/CSS3/JS technologies predominantly drive development within the web environment. To ensure efficient interaction between the frontend and backend, this system adopts a frontend–backend separation approach. The frontend sends Post or GET requests to the backend for business operations and receives returned data in formats such as JSON. The results are then processed and displayed in the corresponding modules.

The data transmission part utilizes WebSocket technology for communication between the frontend and backend. Its main functionality includes sending and receiving hypertext

transfer protocol requests, with data formats being JSON data, form data, and static resources related to the topology. Additionally, to accommodate the diverse and complex nature of data interaction between the frontend and backend, a pre-established agreement on the format and meaning of data exchange is required. This agreement encompasses various aspects, such as behavior, business module types, icon names, specific data values, and more.

In the backend service layer, the focus is on handling the interface communication between the frontend and the algorithms. This involves processing algorithmic data, such as calculating the service capabilities of UAVs, managing system parameters and algorithm-related information, and recording and returning the runtime of multiple algorithms. As a land–air collaborative logistics delivery platform, this system emphasizes frontend visualization. Therefore, the backend is developed using the KOA2 framework. KOA2 has the advantages of supporting async and await operations, as well as the onion model middleware. Therefore, when designing the backend, it is necessary to consider the design of middleware, including but not limited to calculating the total processing time of server requests, adding appropriate content types to response headers for better handling of data returned by the server by the browser, reading the contents of files from specified directories based on URLs, and configuring cross-origin settings.

The business service section primarily includes functions such as topology display, topology configuration, visual display, and dynamic trend display to complete the corresponding business. The logic layer part mainly involves Reliable grOund–Air Delivery (ROAD) parcel algorithms: user allocation algorithm based on reliable delivery, rapid iteration algorithm based on a rich operator, and action selection algorithm based on RL. Among them, the user allocation algorithm based on reliable delivery: network topology and user demands are used as algorithm inputs, and the algorithm generates initial solutions by assigning users to different trips. The design objective of the rapid iteration algorithm based on rich operator is to progressively enhance the current routes during each iteration, gradually reducing the value of the objective function. To ensure the rapid convergence of the solution, a perturbation controller is used to monitor the changes in the solution. The action selection algorithm based on RL is employed to efficiently select operators for the current solution. In this algorithm, the combinations of operators generated in each iteration form an action space, and the operations executed by these operators are lightweight. By inputting the current state, a neural network generates an action probability vector. The network weights are trained using the policy gradient method. This algorithm achieves good performance in various metrics and effectively optimizes UAV schedules to ensure service reliability.

### 3.2.2. System Functional Architecture Design

The functional module plan of the system is as shown in Figure 6. In the context of visualizing reliable delivery logistics with UAV and vehicle collaboration, the system outlined in this paper primarily consists of the following modules: topology parameter management module, route display module, operation result display module, target result display module, service capability display module, and system performance display module.

The topology parameter management module primarily allows system users to customize various parameters of the current topology. This includes the number of users, single/dual objectives of the algorithm, UAV payload weight, UAV maximum power, and the selection of comparative algorithms. By initializing the network topology and configuring the UAV parameters, the algorithm can be executed once confirmed.

The route display module is responsible for frontend rendering based on the topology selected by the user. Before running the algorithm, it visualizes the distribution and demands of the users. Once the algorithm completes, it utilizes the route data returned from the backend to display the paths. These paths encompass both the driving paths of the vehicles and the flight paths of the UAVs.

The operation result display module mainly compares the selected comparative algorithms with the proposed algorithm in terms of runtime and performance. The target result display module primarily showcases the algorithm's performance under single-objective optimization or bi-objective optimization. Its functionality is designed to validate the reduction in runtime achieved by converting bi-objective optimization to single-objective optimization within a reasonable solution range. The service capability display module focuses on defining the service capabilities of the UAVs proposed by the algorithm. Based on the final generated routes by the algorithm, the backend performs data statistics and calculations to display the comparison between the current algorithm and the comparative algorithms in terms of UAV service capabilities. This validation demonstrates that the proposed algorithm in this paper can comprehensively schedule UAV resources and plan UAV battery power efficiently. The final system performance module focuses on the stability and robustness of the algorithm proposed in this paper. It showcases the trend of algorithm results converging to a stable state during the iteration process, ultimately obtaining the optimal solution to the problem. Moreover, in situations where the algorithm may encounter local optima, the proposed perturbation controller can reconstruct and improve the current optimal solution.

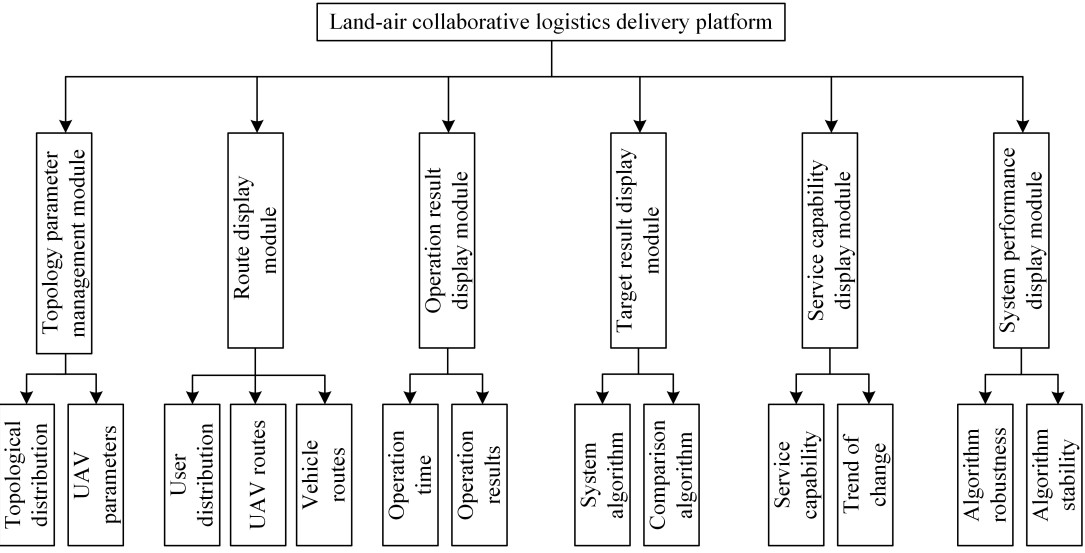

**Figure 6.** System function module planning diagram.

*3.3. System Detailed Design*

3.3.1. Detailed Design of Core Function Modules

(1)   Topology Parameter Management and Route Display Module

   ①   Number of Users in the Network Topology: 20, 30, 50, default 20;
   ②   Light and Heavy Package Distribution: Below 86% by default;
   ③   UAVs Load: 1 kg, 3 kg, 5 kg, default 3 kg;
   ④   UAVs Energy Consumption: 3000 mAh, 4000 mAh, 5000 mAh, default 3000 mAh;
   ⑤   Comparison Algorithms: Joint Trajectory Design and Task Scheduling (TDTS), Collaborative Pareto Ant COlony (CPACO), Greedy-based Wireless Allocation Algorithm (GWAA), Clustering-based Two-layered (CBTL), default select all;
   ⑥   Optimization Goals: Single objective, bi-objective, default single objective.

After the user selects the required algorithm parameters and related settings, the system displays the map with the current network topology and the distribution of user coordinates and demands. Once the basic information is displayed, the parameters are sent to the backend for algorithm invocation. When the algorithm finishes running, the routes and optimized objective values are passed to the backend. The frontend then utilizes the generated routes for UAV flight and vehicle operations. If the user chooses to directly

run the algorithm, it will be invoked with default parameters set by the system. Prompts will be provided for parameter settings and optimization objective function definitions in conjunction with the topology parameter management module. Figure 7 shows the business flow diagram of the land–air collaborative delivery logistics platform, which describes the business logic within each module and the interactions between modules.

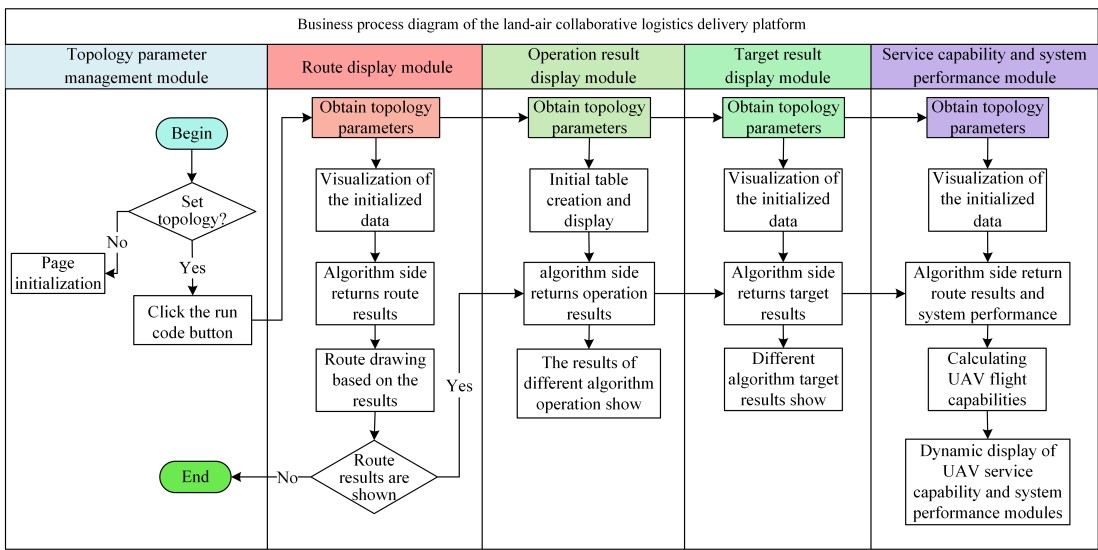

**Figure 7.** Business process diagram of land–air collaborative delivery logistics platform.

(2)  Operation Results Display Module

The frontend of the result display module primarily utilizes Echarts for visualization. After the completion of algorithm execution, the results of multiple algorithms are actively pushed to the frontend for display. The system's runtime sequence diagram is shown in Figure 8. The algorithm results displayed on the frontend page consist of two parts: the results of the proposed algorithm and the results of the comparative algorithms. After the algorithm generates paths and objective values, the backend processes and calculates the data, then returns the results to the frontend for comprehensive display on the interface, including the display of the growth trend of the results.

(3)  Objective Results and Service Capability Display Module

The objective results display module primarily shows the changing trends in optimization objective values for the current algorithm and comparison algorithms across various network topologies and UAV parameters. These trends are visually presented within the module through bar charts. The service capability module mainly calculates the UAVs service capacity for each user based on the current algorithm's output path. By comparing the service capabilities of UAVs during the service process across different algorithms, this module demonstrates the advantages of the algorithm proposed in this paper for UAV scheduling.

(4)  System Performance Display Module

The system performance display module primarily provides real-time visualization of the objective function values generated during each iteration of the algorithm. It presents the stability and robustness of the system in the form of line charts. In cases where multiple iterations yield unchanged values, responsive prompts are displayed to indicate the utilization of a disturbance controller designed within the algorithm. This helps the current optimization value break out of the local optimal in the current problem.

(5)  Frontend and backend Request Modules

The above description primarily outlines how the frontend module visualizes data from the backend. The frontend and backend request module, on the other hand, focuses

mainly on designing the requirements for requests, including frontend-initiated requests and backend's active push to the frontend. This system incorporates the WebSocket request method. WebSocket maintains a long-lasting connection between the browser and the server, enabling real-time communication. With WebSocket, data can be actively pushed from the backend to the frontend, ensuring real-time data transmission. In this system, the implementation of the backend's active push functionality is achieved through creating a WebSocket instance object and setting up event listeners. In the frontend modules, different modules request various types of data. Therefore, during the request process between the frontend and backend, the system establishes a predefined interaction method between the client and server, using JSON format. This format includes properties such as action, socketType, chartName, and value, which represent module information and request details. The action represents a specific behavior. In this system, there are three types of actions: obtaining chart data, initiating a fullscreen event, and changing the theme. In order to uniformly handle data received from the backend, the frontend establishes a global registration function. When specific data are sent from the backend, the corresponding registered function is invoked to deliver the data to the respective module. Therefore, socketType indicates the type of business module, facilitating the execution of callbacks and returning data between the frontend and backend. The chartName field indicates which module is requesting data from the server. The value field represents the specific function value. When retrieving chart data, the frontend can omit the value. However, if there is a need for full-screen events or theme switching, the field can be included.

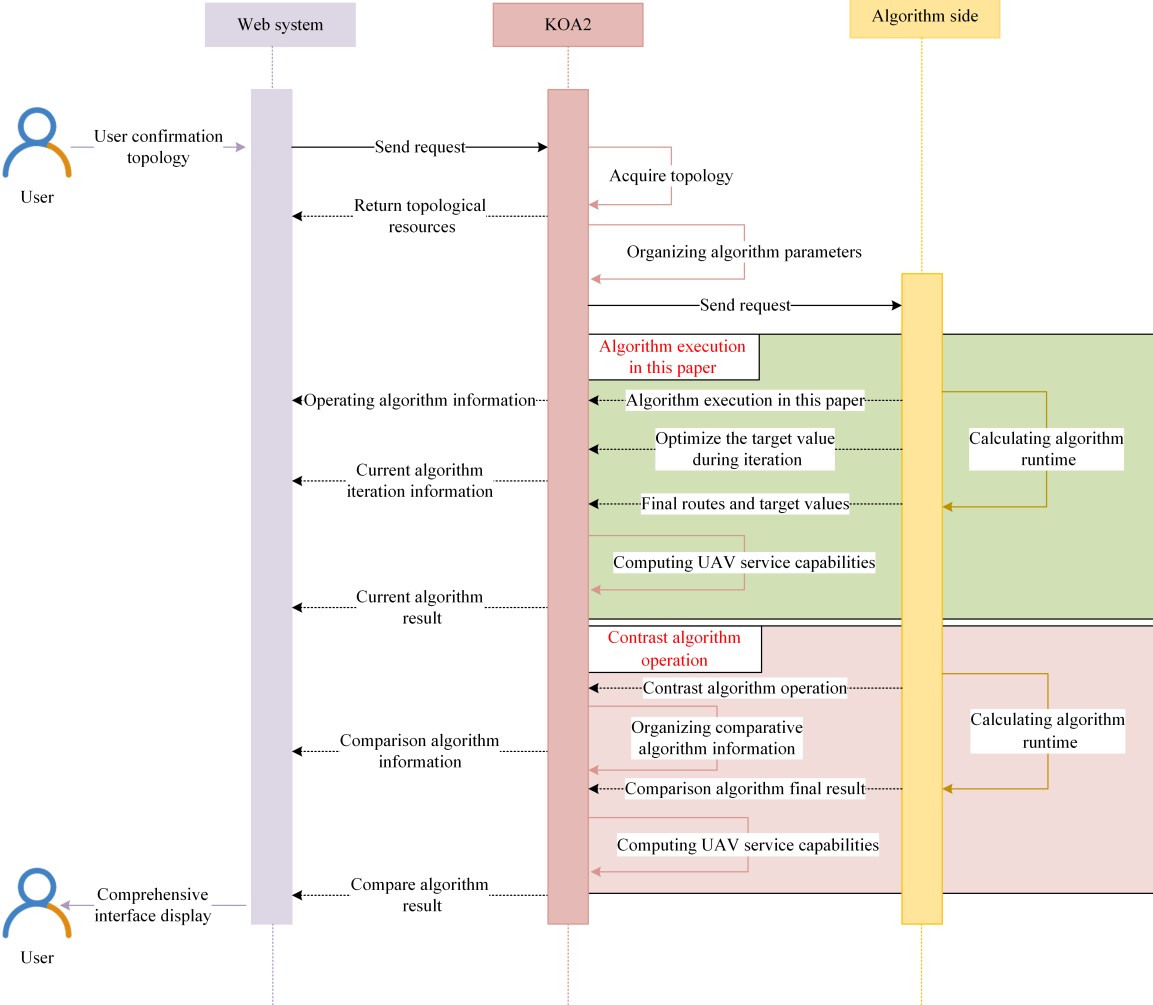

**Figure 8.** Sequence diagram of system operation.

(6)   Frontend and Backend Interaction Modules

The system design includes frontend fault tolerance handling and a reconnect mechanism to ensure the reliability of frontend and backend interactions. The frontend fault tolerance handling primarily addresses the issue of the client and server not being able to immediately establish a successful connection after refreshing the interface. There may be instances where data retrieval functions fail to send successfully while the connection is active. To address this, the system incorporates fault tolerance handling in the request sending module by recording the number of attempts made to send data. As the number of data-sending attempts increases, the delay for the next connection is extended. When the connection is unsuccessful for several times or the initialization is successful but the server is closed, the disconnection and reconnection mechanism is designed. The system keeps track of the number of reconnection attempts and increases the time interval before the next reconnection. After several unsuccessful attempts, a prompt message is sent from the frontend, and the connection is actively closed. Upon refreshing, the connection module is triggered again.

(7)   Backend Module Design

The backend of this system is designed using the Node.js web server framework KOA2. KOA2 module is chosen due to its features of supporting asynchronous operations and middleware in the onion model. The sequence of invocation is as follows: after the request arrives at the server, it goes through the first, second, and third layers of middleware in order. It then returns to the second layer of middleware, where the request undergoes processing for the second time. Finally, it returns to the first layer of middleware, responds to the request, and returns it to the browser. With the help of the server-side framework, this system designs three middleware layers, each with its own functionality. The first layer calculates the total processing time for handling requests on the computation server. The second layer adds the appropriate MIME type to the response headers based on the corresponding content. The third layer retrieves the content from a specified directory based on the URL requested by the frontend. Each layer corresponds to its specific middleware. Additionally, the server sets the response headers and handles cross-origin resource-sharing issues.

3.3.2. Relationship Design between Functional Modules

(1)   Frontend Modules Interact with Backend

Due to the fact that each module in the frontend requests data from the backend or the backend initiates requests to specific modules based on their functional requirements, the system designs the frontend and backend interaction as a singleton pattern. It defines WebSocket to listen for server connections and receive backend information. Additionally, registration functions are defined to invoke corresponding callback functions and relay the data to the designated modules upon retrieval.

(2)   Interaction between Frontend Modules

The interaction between frontend modules mainly focuses on the functional requirements of full-screen switching and theme switching provided by the system. Full-screen switching allows users to focus on a specific module and enlarge it for better visualization. The implementation of this feature involves layout adjustments, enabling full-screen mode, and sending data related to full-screen events. Theme switching, on the other hand, provides the functionality to change the system's appearance. Its implementation includes storing the current theme data, handling click actions to switch themes, listening for theme changes, and adapting native HTML styles accordingly.

In the theme switching feature, the relevant information of the current theme needs to be used in multiple components. Therefore, this system uses VueX to add a data store theme and performs corresponding theme switching operations using mutations. By using property watchers to monitor theme changes, the system enables theme switching. In the

process of adapting to native HTML theme styles, the system defines and exports the style data and functions that need to be switched under two themes, making it convenient for modules to access theme-related information. Each module is equipped with the respective functions to handle the theme switching.

## 4. System Implementation

After completing the preliminary stages of requirement analysis and system design, we proceed to the coding and implementation phase of the system. System implementation is a crucial stage in software development as it directly determines the final quality of the software and lays the foundation for subsequent testing and maintenance. This section will explain the implementation of some key functionalities of the system by showcasing relevant code snippets and system interfaces.

### 4.1. System Development Environment Construction

The main development tools for this system include PyCharm, Visual Studio, etc. As shown in Table 2:

**Table 2.** System software configuration.

| Software Configuration | Software Description |
| --- | --- |
| Operating System | MacOs 13.2.1 |
| Visual Studio | Visual Studio 1.76.0 |
| Pycharm | Pycharm 2020 |

### 4.2. Core Functional Modules Implementation

(1)　Topology Parameter Management and Route Display Module

In the parameter management module of this system, users can initialize the network topology by single selection, multiple selection, and sliding bar. They can initiate algorithm running request to the backend by running code button and have a restore default button to restore the default network topology by button click. When the user clicks the run code button, it sends a request to the backend. At the same time, it initiates requests to the path topology module and algorithm robustness module. The path topology module constructs an appropriate topology based on the user's selection, while the algorithm robustness module starts a listener function to dynamically display iterative values from the backend algorithm. In the system, the module interaction and communication between the frontend and backend WebSocket are designed using the singleton pattern. Due to the similarity in request initiation during the module interaction process, the system designs identical callback functions and stores them uniformly. The network topology parameter module is illustrated in Figure 9.

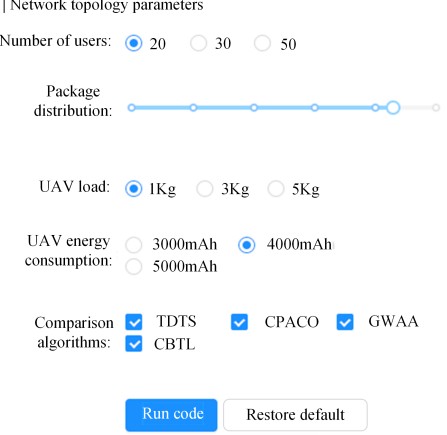

**Figure 9.** Module diagram of network topology parameter management.

After the user clicks the run code button, the system first displays the user distribution nodes in the network topology based on the user-defined number of users, as shown in Figure 10.

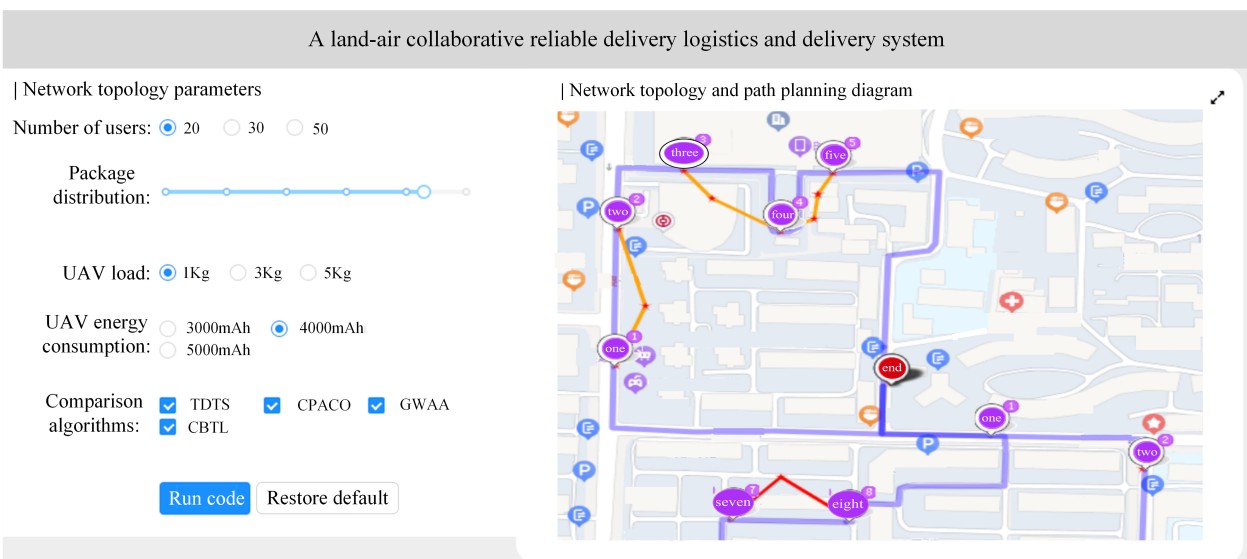

**Figure 10.** Module diagram of user distribution.

Once the system algorithm finishes running, the frontend displays the flight paths of UAVs and vehicles based on the route nodes returned by the backend. The results are shown in Figure 11.

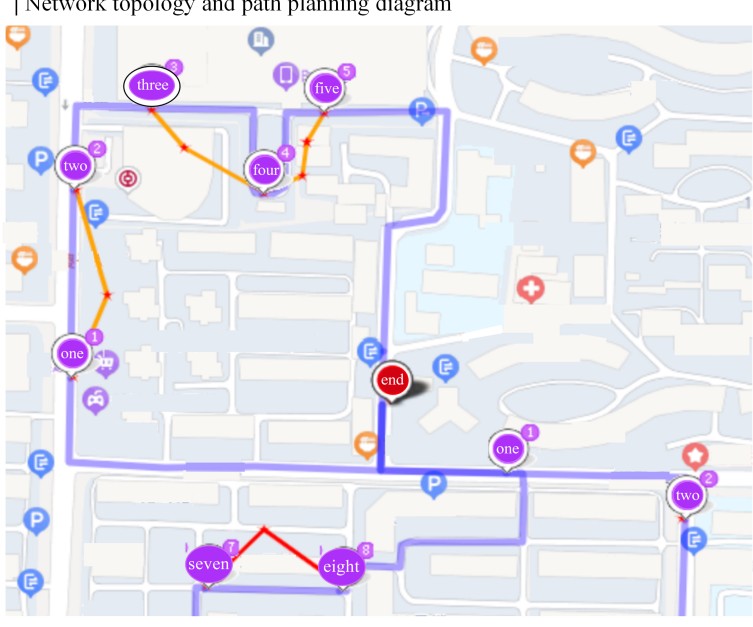

**Figure 11.** Module diagram of network topology and path planning.

(2)   System Performance Display Module

During the development process, this system utilizes timers to continuously add data to the line chart. As the performance display module requires real-time visualization by constantly listening to the data sent from the backend, the development effect of this module is shown in Figure 12.

(3)   Backend Middleware Design Modules

The backend of this system employs middleware to invoke algorithms and respond to data.

(4)    Full Screen Switch Function Implementation

Due to the presence of multiple modules on the system display page, to meet the user's demand for detailed browsing of a specific module, the system incorporates a full-screen switching feature.

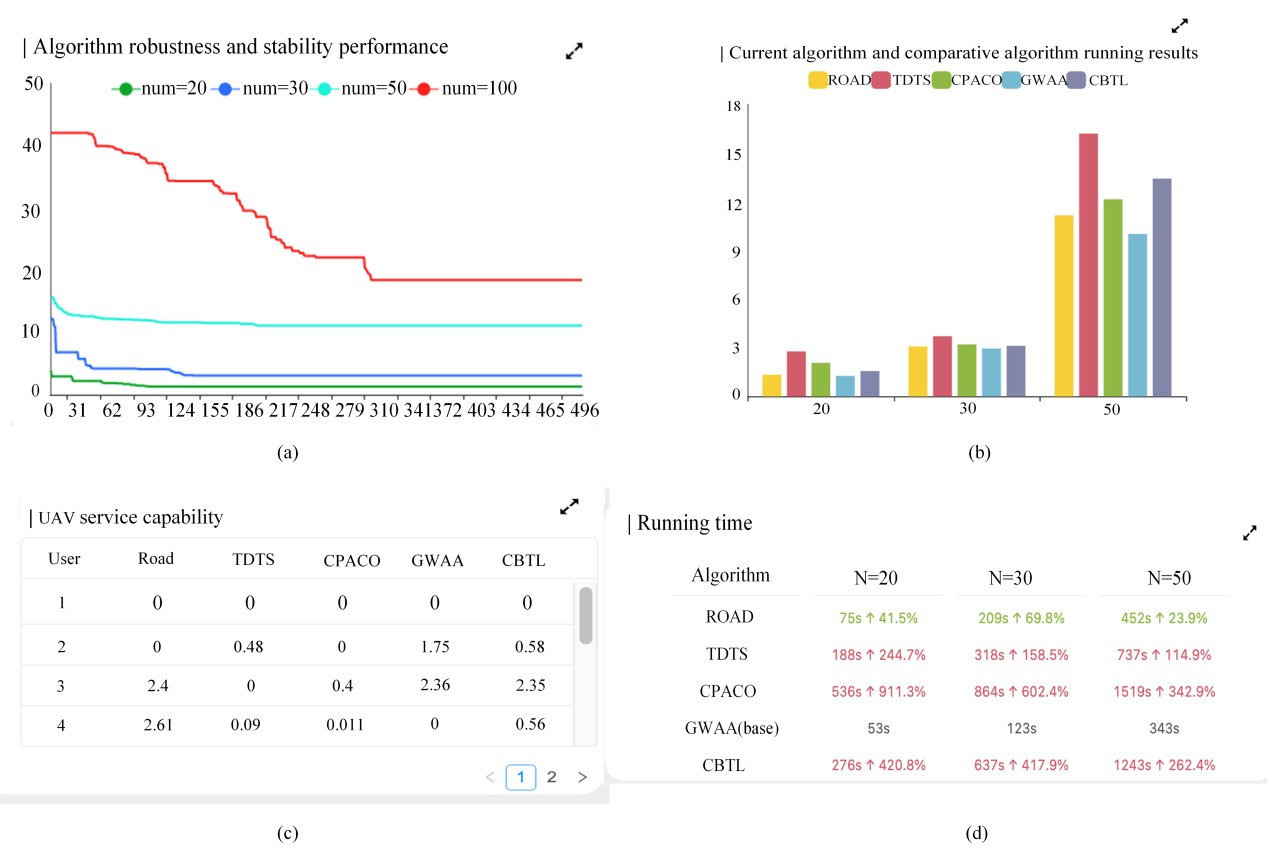

**Figure 12.** Module diagram of system performance demonstration: (**a**) Algorithm robustness and stability module. (**b**) Compare the algorithm results module. (**c**) UAV service capability module. (**d**) Runtime results module for different algorithms.

## 5. System Testing

System testing is an essential phase in software development that directly ensures the smooth operation of the system after its deployment. Through system testing, it becomes possible to effectively determine whether the designed system in this paper functions properly and meets the expected requirements. Software testing, as an important component of software engineering, verifies the usability, practicality, and robustness of the system software through manual or automated methods.

Figure 13 shows the overall functionality display of the system. It is composed of various visualization modules mentioned above and supports features such as theme switching and full-screen events. The performance results of the proposed algorithm and other comparative algorithms under different factors are presented using bar charts, tables, line graphs, and other forms.

This paper conducted functional testing on the land–air collaborative logistics delivery platform and formulated corresponding system test materials. These materials encompass the network topology parameter module, route planning module, comparative algorithm module, service capacity and runtime module, as well as specific test scenarios. The detailed description is presented in Table 3.

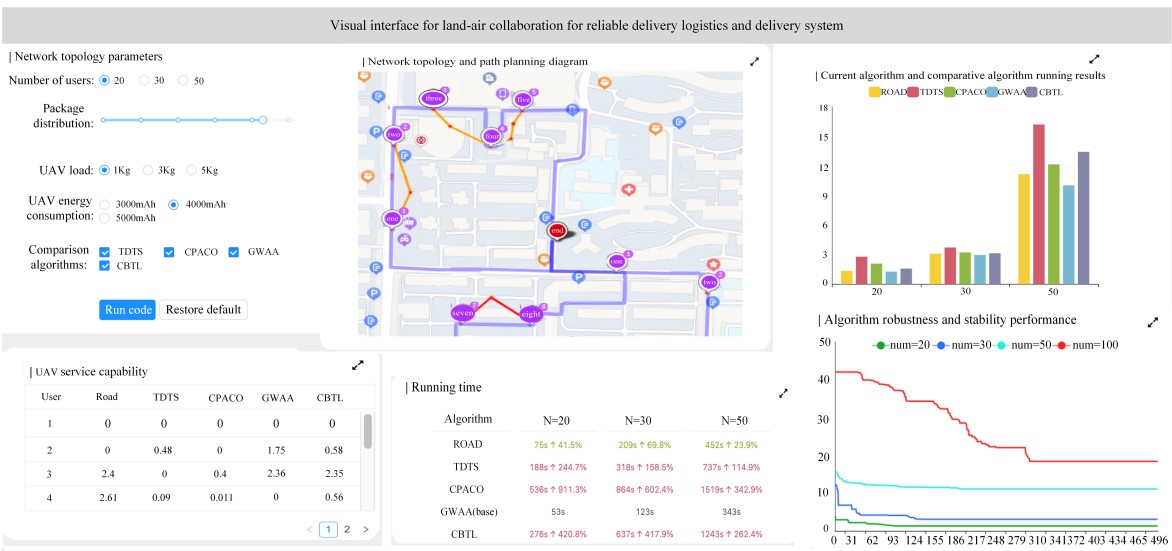

**Figure 13.** Display diagram of the overall function module of the system.

**Table 3.** System of test content.

| System Testing Specifics |
|---|
| Network Topology Parameters Module: whether the triggering algorithm and reset function are functioning properly.<br>Route Planning Module: whether the route can be displayed correctly according to the algorithm output.<br>Comparison Algorithm Module: whether to display according to the user's choice of comparison algorithm.<br>Service Capability, Running Time, Algorithm Stability Module: whether it can display correctly according to the returned data. |

As per the testing requirements, this system undergoes validation using diverse test cases, encompassing various user actions and the presentation of algorithm scheduling results before and after. The comprehensiveness of the system is assessed by scrutinizing the sequence of steps, anticipated outcomes, actual results, and pass/fail status. The experimental test results are shown in Table 4.

**Table 4.** Table of test cases and results comparison.

| Use Case | Step | Expected Result | Test Result | Whether to Pass |
|---|---|---|---|---|
| 01 | After the user modifies the parameter configuration, click run | The system runs algorithm; topology module shows user distribution. The algorithm robustness module displays the results of each iteration | Same as expected | Yes |
| 02 | User clicks on the restore defaults button | Parameter module restores default values | Same as expected | Yes |
| 03 | After the algorithm finishes running, the route is automatically displayed. | The topology module automatically displays UAV and vehicle travel routes | Same as expected | Yes |
| 04 | Automatic display of system algorithm and comparison algorithm results after the algorithm finishes running | Display comparison algorithms and their related data based on user selection | Same as expected | Yes |
| 05 | Automatic calculation of UAV service capability after algorithm run | During the service process, the service capability of UAVs is calculated for each user and displayed in tabular form | Same as expected | Yes |
| 06 | Automatic display of different algorithm run times after the algorithm finishes running | Displaying the runtime of different algorithms based on user selection and calculating the time trend in comparison to the baseline algorithm | Same as expected | Yes |
| 07 | Click on the system skinning button | Change between two themes | Same as expected | Yes |
| 08 | Click on the full-screen view button | Full-screen display of the corresponding module | Same as expected | Yes |

## 6. Conclusions

This study delves into a reliable delivery logistics system based on the collaboration between UAVs and vehicles. It analyzes the shortcomings of traditional logistics delivery services and designs a land–air collaborative logistics delivery platform. The platform utilizes blockchain technology to guarantee the security and accuracy of the data. Moreover, the system is primarily employed for logistics delivery within medium and large cities. The feasibility of this system hinges on reducing drone costs and enhancing their capabilities, enabling swift parcel delivery within confined areas. For extensive parcel distribution, vehicle management governs the main delivery routes, while UAVs will be dedicated to executing the "last-mile" delivery segment. Through the synergy of their services and efficient resource management, this collaborative model significantly reduces delivery times and enhances overall logistics efficiency.

In conclusion, the system designed in this paper has brought innovative solutions to the logistics and delivery industry. In the future, our work aims to address the challenges posed by the dynamic growth in user demands and the varying capabilities of UAVs and vehicles in the context of collaborative logistics delivery. We will continue our research to handle the dynamic addition of network user nodes and make appropriate adjustments to service routes, ensuring high-quality user services while promoting energy conservation and emissions reduction.

**Author Contributions:** Conceptualization, H.L. and W.W.; methodology, H.L., S.Z. and Z.L.; software, H.L., S.Z. and Z.L.; validation, Z.L., A.T. and W.W.; formal analysis, H.L. and S.Z.; investigation, H.L., S.Z. and W.W.; resources, H.L. and W.W.; data curation, H.L., S.Z. and A.T.; writing—original draft preparation, H.L.; writing—review and editing, W.W.; visualization, Z.L. and S.Z.; supervision, H.L., S.Z. and A.T.; project administration, H.L., S.Z. and W.W.; funding acquisition, A.T. All authors have read and agreed to the published version of the manuscript.

**Funding:** This work was funded by the Researchers Supporting Project number (RSPD2023R681), King Saud University, Riyadh, Saudi Arabia.

**Institutional Review Board Statement:** Not applicable.

**Informed Consent Statement:** Not applicable.

**Data Availability Statement:** Not applicable.

**Conflicts of Interest:** The authors declare no conflict of interest.

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
