# Peer review of "A Reliable Delivery Logistics System Based on the Collaboration of UAVs and Vehicles"

_sustainability, doi:10.3390/su151712720_

Round 1

Reviewer 1 Report

this paper develops a land-air collaborative reliable delivery logistics and delivery system, showcasing the routes of both vehicles and UAVs. This mode of collaboration not only reduces the operational costs associated with traditional logistics delivery but also achieves energy savings and emission reduction goals. 

1- I think the summary is too long and needs to be revised. It's hard to see the main contribution.

2- There are a number of errors in the English, and it is highly recommended that you revise the text.

3- Figure 1 needs to be better explained in the text.

4- I find that your model is too simplified and needs to be improved. Furthermore, the results are not compared. In terms of comparison with the literature, I suggest that you add a table comparing your work with the most important articles in your research.

5- please add this two papers:

M. A. Ouamri, R. Alkanhel, D. Singh, E. M. El-kenaway and S. S. M. Ghoneim, "Double deep q-network method for energy efficiency and throughput in a uav-assisted terrestrial network," Computer Systems Science and Engineering, vol. 46, no.1, pp. 73–92, 2023.

Ouamri, M.A., Singh, D., Muthanna, M.A. et al. Performance analysis of UAV multiple antenna-assisted small cell network with clustered users. Wireless Netw 29, 1859–1872 (2023). https://doi.org/10.1007/s11276-023-03240-9

Extensive english is required 

Reviewer 2 Report

- the theoretical background of the paper is definitely too narrow; there is a significant lack of important terms and papers which should be explained and added to the reference list; some of them are here: https://doi.org/10.3390/en15134668 https://doi.org/10.14716%2Fijtech.v7i4.2578 https://doi.org/10.1016/j.giq.2017.11.008 https://doi.org/10.1016/j.procs.2021.08.222 - please add maps in the English language in Figures 10, 11, 13, - what are the limitations of the proposed system? Please describe, - what are the real possibilities of implementation of the system and where it can be done? Please describe, - what are the directions of development of the proposed system? Please describe

Reviewer 3 Report

1: In what ways is the new delivery logistics system proposed in your article an improvement over the current system? You can summarize the contribution points based on the literature review

2: Your article gives a very detailed description of the system requirements and system architecture, showing its functionality, but it seems to lack a description of its application area, what kind of application object is the system oriented to? What kind of application is the system aimed at? Is it the current city?

3: You mentioned that the system has embedded a variety of algorithms for users to choose from, but the description of these algorithms is not comprehensive, what is the application scope of these algorithms? What is the area of application of these algorithms? How should users choose them?

4: Your experimental design seems to prove that the system is usable, but how do you prove experimentally that it is state-of-the-art?

5: You summarize the system's effectiveness at the end of the first section, but how are these benefits achieved? There seems to be a lack of corresponding experimental proof, e.g. carbon emission reduction, system robustness, etc.

Moderate editing of English language required

Round 2

Reviewer 2 Report

Dear Authors,

you answered well my questions, but you did not present your answers in the paper - please add your answers to the manuscript.

Reviewer 3 Report

Please check a little bit about the format of the pictures and formulas again, and modify some English usage and mistakes.

Minor editing of English language required
